# Airway and Respiratory Devices in the Prevention of Ventilator-Associated Pneumonia

**DOI:** 10.3390/medicina59020199

**Published:** 2023-01-19

**Authors:** Luis Coelho, Patricia Moniz, Gonçalo Guerreiro, Pedro Póvoa

**Affiliations:** 1NOVA Medical School, New University of Lisbon, 1300 Lisbon, Portugal; 2Department of Intensive Care, Hospital de São Francisco Xavier, CHLO, 1449 Lisbon, Portugal; 3Public Health Department, Pulmology Diagnostic Center Dr. Ribeiro Sanches, Regional Health Authority for Lisbon and Tagus Valley, 1700 Lisbon, Portugal; 4Center for Clinical Epidemiology and Research Unit of Clinical Epidemiology, OUH Odense University Hospital, 5000 Odense, Denmark

**Keywords:** ventilator-associated pneumonia, airway devices, respiratory devices, prevention, mortality

## Abstract

Ventilator-associated pneumonia (VAP) is the most common ICU-acquired infection among patients under mechanical ventilation (MV). It may occur in up to 50% of mechanically ventilated patients and is associated with an increased duration of MV, antibiotic consumption, increased morbidity, and mortality. VAP prevention is a multifaceted priority of the intensive care team. The use of specialized artificial airways and other devices can have an impact on the prevention of VAP. However, these devices can also have adverse effects, and aspects of their efficacy in the prevention of VAP are still a matter of debate. This article provides a narrative review of how different airway and respiratory devices may help to reduce the incidence of VAP.

## 1. Introduction

Ventilator-associated pneumonia (VAP) results from the infection of pulmonary parenchyma in patients under mechanical ventilation (MV) for more than 48 h. It is a very common ICU-acquired infection, occurring in up to 50% of mechanically ventilated patients, and is associated with an increased duration of MV, high antibiotic consumption, increased morbidity, and mortality [1,2,3].

The protection of the lungs from infections depends on a set of defense mechanisms that prevent the invasion of the lower airways by pathogenic microorganisms. Anatomical barriers (larynx and glottis), cough, mucociliary clearance, and local humoral and cellular immunity are the main defenses that prevent the onset of serious infections such as pneumonia [4]. Whenever it is necessary to perform intubation, regardless of the clinical condition, these defense mechanisms are suppressed and the access of potentially pathogenic microorganisms to the lower airways is facilitated. Aspiration of pathogens can occur when performing orotracheal intubation, invasive procedures such as bronchoscopy, or aspiration of tracheal aspirations by translocation during orotracheal intubation or through the passage of supraglottic secretions through the folds of the endotracheal tube (ETT) cuff. The bacteria that cause VAP are usually from the oropharynx, or less frequently, from the digestive tract [5]. Usually, the microorganisms involved in VAP that occur in the first days of mechanical ventilation are those that constitute the normal flora of the oropharynx. However, with increasing ventilation time, the probability of infections by resistant bacterial agents increases [6]. The agents most frequently involved are Gram-negative Pseudomonas aeruginosa, Escherichia coli, Klebsiella pneumoniae, and Acinetobacter species, and Gram-positive Staphylococcus aureus. The main risk factors for the appearance of multidrug-resistant agents are the prior colonization or infection with MDR pathogens, ARDS preceding VAP, acute renal replacement therapy prior to VAP, previous exposure to antibiotics, and the presence of septic shock at the time of VAP [7].

As a result, the prevention of VAP is a fundamental strategy to reduce its incidence and impact on the morbidity and mortality of ventilated critically ill patients [8]. The reduction in the incidence of VAP has been achieved with practices that reduce the need for mechanical ventilation in patients with acute respiratory failure, such as noninvasive ventilation or high-flow oxygen therapy, or that reduce the duration of mechanical ventilation, such as light sedation and early mobilization and extubation [9]. In many observational studies, the combination of these strategies has been shown to positively influence mortality, although their true impact still needs to be validated by randomized studies with less risk of bias [10].

This article provides a narrative review of how different airway and respiratory devices may help to reduce the incidence of VAP.

## 2. Subglottic Secretion Drainage

The microaspiration of secretions containing bacterial pathogens into the lower respiratory tract constitutes the main mechanism for VAP occurrence [11]. Impaired laryngeal function caused by the ETT, regurgitation from gastroesophageal sphincter dysfunction, reduced upper airway reflexes, supine positioning, and enteral feeding are factors that increase the risk of microaspiration in patients under invasive MV [12]. ETT cuffs may prevent macroscopic aspiration but are unable to protect against local microaspiration [13].

First introduced in 1992, subglottic secretion drainage (SSD) constitutes a recommended nonpharmacological prevention strategy for VAP [11,14]. This measure reduces the incidence of microaspirations with the use of a specific ETT possessing an accessory channel for subglottic secretion drainage. This channel can be connected to either intermittent or continuous suction, at rates varying from −20 mmHg to −150 mmHg [11,15].

Although the use of SSD is not yet widespread, probably due to poor strength of evidence on certain outcomes, recently updated guidelines on preventive VAP strategies recommend the use of SSD. The 2022 American practice recommendations of the Society for Healthcare Epidemiology in collaboration with the Infectious Diseases Society of America reclassified SSD from an essential practice to an additional approach to VAP prevention [16]. This is mainly due to insufficient data regarding the impact on ICU mortality, length of stay, and duration of MV. Nevertheless, SSD has been proven to reduce the incidence of VAP and may reduce the duration of MV in patients requiring more than 48–72 h of endotracheal intubation. Endotracheal tubes with SSD ports should therefore be considered in such patients [16]. Some European guidelines recommend SSD every 6 to 8 h as part of a multimodal hospital-acquired pneumonia/ventilator-associated pneumonia (HAP/VAP) prevention approach in the ICU, while others highly recommend SSD as a VAP prevention bundle [17,18].

Subglottic secretion drainage has been the most frequently studied preventive measure for VAP, with multiple randomized controlled trials and meta-analyses demonstrating reductions in VAP incidence [19,20]. Caroff et al. recently conducted a systematic review and meta-analysis to evaluate the impact of SSD on the duration of MV, ICU and hospital length of stay, ventilator-associated events, antibiotic use, stridor, reintubations, and mortality. Seventeen RCTs comparing SSD versus no subglottic secretion drainage in adult patients on mechanical ventilation were included with a total of 3369 patients. Although SSD was associated with significant reductions in VAP with an overall RR of 0.58 (95% CI, 0.51–0.67; *p* < 0.0001), no improvement in duration of MV, ICU length of stay or mortality (ICU mortality RR, 0.94; 95% CI, 0.82–1.08; *p* = 0.38) was shown [19].

Mao et al., in a recent meta-analysis, evaluated the effects of SSD on VAP incidence as a primary outcome and its impact on the following secondary outcomes: duration of MV, the time-to-onset of VAP, Gram-positive or Gram-negative bacteria causing VAP, the incidence of early-onset and late-onset VAP, ICU and hospital mortality, ICU length of stay, reintubation, and incidence tracheotomy. Overall, in all included trials, SSD reduced the risk of VAP. Upon analysis of only high-quality studies with 901 participants for the analysis of VAP incidence, the RR for SSD versus no SSD was 0.54 (95% CI 0.40–0.74; *p* < 0.00001), and the number needing treatment was 10.49. Regarding secondary outcomes, SSD reduced the incidence of early-onset VAP, duration of mechanical ventilation, and gram-positive or Gram-negative bacteria. Time-to-onset of VAP was also delayed. No difference was found between groups regarding the other secondary outcomes [20].

A recently updated meta-analysis including 20 RCTs comprised 3684 ICU patients undergoing MV. This systematic review assessed the effectiveness of SSD for the prevention of VAP and the improvement in outcomes such as mortality, MV duration, ICU, and hospital length of stay, outcomes that have not yet been clearly affected by SSD. This study was the first to find decreased mortality associated with SSD use. Although the duration of MV and ICU/hospital length of stay were not improved, SSD remained effective in reducing VAP incidence [21]. However, the authors posteriorly published a correction to the original paper stating that errors had since been detected. One of the most important alterations was due to one RCT being mistakenly accounted for twice, once for hospital mortality and once for ICU mortality [22,23]. The results after adequate data revision were mainly affected by the new weight held by the Damas et al. study (original weight: 31.0%, corrected weight: 20.7%). Mortality results were modified (RR 0.92, 95% CI 0.83–1.02, *p* = 0.098), with a nonsignificant trend of lower mortality rates in patients randomized to SSD. These new results were in line with previous meta-analyses that had not found an impact on mortality. A reduction in VAP incidence among patients undergoing SSD remained [22].

The risk of mucosal damage secondary to aspiration is increased due to the higher external diameters of ETTs used for SSD compared to conventional ETT, or if the suction pressure is too negative. Although concerns have been raised regarding complications associated with SSD implementation, namely mucosal injuries [24], recent studies consider this procedure safe. Vallés et al. in a recent prospective observational study of ICU patients submitted to continuous SSD, recorded clinical airway complications during the postextubation period, where subglottic and tracheal lesions were classified with the use of computed tomography (CT) of the neck. The study population of 86 patients was analyzed, with 15.1% considered difficult intubations. There was no difference between patients with lesions detected on CT (23.4%) and those without regarding the length of ICU stay, airway and intubation variables, age, or APACHE II score. The risk of significant complications such as extubation failure was 8.1% in patients undergoing continuous SSD, and lesions observed on CT were not severe. Overall, the procedure was considered safe, and the incidence of lesions in patients undergoing SSD was not higher than that previously reported in intubated patients without SSD [25].

## 3. Cuff Shape and Material

Tracheal cuff shape could play an important role in the prevention or reduction in VAP [26]. Modifications are usually designed to eliminate folds in the inflated cuff that would otherwise allow microaspiration through the folds. Although several cuff shape modifications are commercially available, there is insufficient clinical data about most of them and their benefits [27].

The tapered-shaped cuff has been the most quoted modification in literature. Theoretically, this shape ensures that the cuff and trachea share the same diameter, providing a permanent sealing zone between the cuff and the tracheal wall [26]. Indeed, some in vitro and animal studies have found a decreased leakage around the cuff as opposed to a standard barrel-shaped cuff. However, these experiments were performed on inanimate objects or heavily sedated animals, eliminating some important factors such as endotracheal tube and patient movement or irregular shape of the trachea, which were previously reported as risk factors for leakage and microaspiration [28,29,30].

Three RCTs found significantly less microaspiration around tapered cuffs in short-term clinical settings [29,30]. Nonetheless, these trials investigated dye leakage shortly after instillation and, therefore, deductions about long-term ventilated patients could not be made. Cuff underinflation, patient movement, and accumulation of subglottic secretions all influence the risk of aspiration and are more likely to occur with an increased duration of MV [28].

A recent meta-analysis of six randomized controlled clinical trials with 1324 patients showed no significant difference in HAP/VAP incidence per patient when tapered cuffs were compared with standard cuffs (odds ratio, 0.97; 95% CI, [0.73–1.28]; *p* = 0.81). Likewise, secondary outcomes such as mortality, length of stay, or duration of mechanical ventilation were similar. This meta-analysis presented some limitations including limited data from unpublished trials, clinical heterogeneity (population and setting differences (ICU vs. postoperative), diagnostic criteria differences, and duration of MV), and methodological weaknesses (lack of blinding and poor reporting of methodology and clinical settings in some studies) [29]. In conclusion, the clinical data did not support the routine use of tapered cuffs for the prevention of VAP. Nevertheless, the use of other shapes and tapered cuffs in addition to the use of SSD and/or continuous cuff pressure regulation remains largely unexplored [29].

As previously noted, the thickness of the cuff contributes to the magnitude of the folds created by it. A thinner material could theoretically reduce channel sizes between the tracheal cuff and the tracheal wall [26,31].

Polyurethane cuffs (PUC) are 40-fold thinner than regular polyvinyl chloride cuffs (PVC) [32]. In vitro studies suggest that PUC might reduce the microaspiration of contaminated secretions and, in an artificial tracheal model, the PUC did not show any folds or channels along the tracheal wall through computed tomography analysis, whereas the standard PVC cuff had folds clearly visible when inflated to the recommended 20 cm of H_2_O [13].

A recent systemic review by Blot et al. compared PUC cuffs to regular cuffs: nine in vitro experiments, one in vivo (animal) experiment, and five clinical studies were included; the primary outcomes demonstrated less leakage or at least a tendency toward better sealing. However, evidence showing lower VAP rates was less robust. Some studies showed no difference in this secondary outcome, probably because microaspiration was postponed rather than eliminated, favoring PUC for high-risk patients with a predicted low ventilation period. This review had its limitations: tube characteristics other than the material may have played a role in sealing efficacy, and study designs and experimental setups and endpoints were notably different. Physical and chemical features of PUC also favor condensation in the pilot balloon, precluding inaccurate cuff pressure measurements [32].

Further evidence is needed before PUC ETTs can be recommended as a widespread VAP prevention [27], and the financial impact of these new-design tubes compared with the extrapolated saving has yet to be analyzed [31].

## 4. Continuous Cuff Pressure Control

Continuous cuff pressure control (CCPC) is an automated feature of ventilators or an external device that allows the constant monitoring and regulation of cuff pressure between a defined range, as opposed to intermittent control, dependent on manual control with a manometer by the staff.

Cuff pressure should be set to balance risks of short-term complications; guidelines recommend maintaining a cuff pressure between 20 and 30 cm H_2_O, as underinflation has been reported as a risk factor for VAP, whereas pressures higher than 30 cm H_2_O may impair tracheal mucosa blood flow leading to tracheal damage, ischemia, and tracheomalacia [33,34].

Even with a properly inflated, standard high-volume, low-pressure cuff, longitudinal folds are created allowing leakage of secretions into the respiratory tree, facilitated by patient and ETT movements such as coughing and cyclic manual cuff pressure checks [28,29].

Continuous proper cuff pressure was theorized to reduce this microaspiration; however, despite the rigorous intermittent manual control of cuff pressure, a higher number of underinflated cuffs have been reported compared with automated CCPC due to unceasing loss of pressure. Valencia et al. assessed the efficacy of automated CCPC in optimizing cuff pressures and preventing VAP. Although they did not demonstrate a statistically significant difference in the VAP rate, cuff pressures were maintained within the target range (20–30 cm H_2_O) in approximately 80% of the 142 patients. Only 0.7% of the CCPC-group patients had cuff pressures less than 20 cm H_2_O compared with approximately 45.3% in the control group, where pressure was monitored every 8 h [35].

The risk of VAP has also been assessed in a recent meta-analysis by Maertens et al. that included eleven RCTs composed of 2092 adult intubated patients. The use of CCPC was associated with a reduced risk of VAP (OR 0.51) and the durations of MV (mean difference −1.07 d) and ICU stay (mean difference −3.41 d). However, mortality was not affected. This conclusion was based on very low certainty of the evidence due to concerns related to blinding, different devices used, and commercial conflicts [36]. New studies are necessary to define the role of these devices in the prevention of VAP.

## 5. Low-Volume Low-Pressure Cuff

Regular high-volume low-pressure cuffs (HVLP) were designed so that the external diameter of a fully inflated cuff exceeds the diameter of the tracheal lumen by 1.5–2 times. A partially inflated HVLP cuff will allow cuff contact with the tracheal mucosa and reflect the low exerted pressure upon the trachea. However, it will cause cuff folds, proportional to the diameter and thickness of the cuff [31].

The use of tracheal tubes with low-volume low-pressure (LVLP) cuffs was suggested to reduce microaspiration and VAP by reducing the size and number of folds.

Several small clinical trials reported improved sealing and lower VAP rates in patients intubated with these tubes [26]. Young et al. specifically compared HVLP to LVLP cuff dye leakage in vitro and in vivo: in a tracheal model, LVLP showed 0% dye leakage. In anesthetized patients, leakage was seen in 5% of the LVLP group compared with 67% of the HVLP group. The group concluded that LVLP cuffs reduce microaspiration. Nonetheless, the study sample was small (38 patients) and so was the ventilation period (during surgery procedures) [37].

Currently, LVLP cuffs have been designed with associated strategies to minimize microaspiration: SSD ports, a tracheal seal monitor, or a coated tube lumen. Although this approach appears to be effective in preventing short-term microaspiration, there are no published data on the incidence of VAP, and further clinical studies are required to better evaluate its utility and safety [27].

## 6. Coated Endotracheal Tubes and Mucus Shaver

After intubation, ETT is rapidly colonized by multiple species of bacteria forming a biofilm. This biofilm develops quickly into a structure that facilitates bacterial and fungi proliferation, making it difficult for antibiotics to penetrate, and facilitating the occurrence of VAP [4].

For this reason, different coatings with antimicrobial properties that reduce colonization and biofilm formation in endotracheal tubes have been tested. The most studied coating and the only one to be evaluated in clinical studies was the silver-based coating, due to its antimicrobial, antiadhesive, and nontoxic properties [38].

Preliminary studies in cardiac surgical patients requiring mechanical ventilation between 12 and 24 h showed that silver-coated ETTs had less bacterial colonization and less accumulation of biological material on their internal surface [39]. However, the duration of MV was too short and, therefore, it was not possible to assess the impact on the incidence of VAP.

Subsequently, a randomized study of more than 1500 patients showed that patients ventilated for more than 24 h with silver-coated ETTs had lower VAP rates than patients with uncoated tubes (4.8% vs. 7.5%; *p* = 0.03). However, this beneficial effect was only evident for the first 10 days of mechanical ventilation, and another limitation of the study was the fact that the control group had a significantly higher number of patients with chronic obstructive pulmonary disease, which is considered a risk factor for VAP [40].

Despite being promising, the effectiveness of coated ETTs decreases over the time of MV, eventually allowing bacterial colonization and biofilm formation and therefore losing effectiveness in preventing VA-LRTI. Still, its usefulness in preventing these infections in patients ventilated for less than 10 days should be considered, as evidenced in a more recent review, albeit with limited data [41].

The Mucus Shaver is a concentric inflatable catheter that allows the removal of mucus and secretions from inside the ETT, preventing its obstruction and hindering the formation of biofilm [42]. Its effectiveness in preventing VAP was also evaluated recently; however, it did not show a significant impact on ETT colonization or in reducing the incidence of VAP [43].

## 7. Heat and Moisture Exchangers

Heat and moisture exchangers (HME) recycle the heat and humidification of exhaled air of the patient in the ventilator circuit, eliminating the need for actively heated humidification during MV. In this way, they prevent condensation in the ventilator circuits and their bacterial colonization, which can therefore contribute to a decrease in the incidence of VAP [31].

The effect of HME versus actively heated humidification in preventing VAP has been evaluated in several clinical studies that have suggested some advantages in its use [44]. Furthermore, its cost seems to be advantageous since it is not necessary to replace the HME daily to maintain its effectiveness or to prevent infection, and it can be used for periods longer than 48 h [45].

However, a later meta-analysis evaluating 12 randomized controlled trials showed that HME and actively heated humidification had a similar VAP rate (relative risk 0.85, 95% CI 0.62–1.16). It was also noted that in the analysis of patients with >7 days of mechanical ventilation, no differences were found in the incidence of VAP [46]. Additionally, HME is associated with the increased viscosity of bronchial secretions, which facilitate episodes of atelectasis, increasing dead space and therefore reducing CO_2_ clearance [46]. Therefore, its use as a means of preventing VAP is not recommended.

## 8. Ventilator Circuit Change

The role of the ventilator circuit in increasing the risk of VAP has been thoroughly discussed. Bacterial colonization of the circuit originates primarily from the patient’s bronchial secretions, and the occurrence of VAP is determined by the risk of accidental backflow of contaminated condensate from the tube into the patient’s airways during circuit exchange procedures. Craven et al. demonstrated that changing the ventilator circuit every 24 h instead of every 48 h increases the risk of VAP [47] and, later, these data were confirmed in a meta-analysis of 10 studies, where it was demonstrated that changing the ventilator circuit every 2 or 7 days has a higher risk of VAP compared to unscheduled changing (odds ratio 1.13; 95 % CI, 0.79–1.6) [48]. In conclusion, the programmed change of the ventilator circuit to prevent VAP is not recommended.

## 9. Closed Tracheal Suctioning System

Tracheal aspiration is essential to avoid the accumulation of secretions in the bronchial tree and prevent ETT occlusion. To perform a tracheal aspiration, it is usually necessary to disconnect the patient from the ventilator circuit, which could result in contamination of the airways. Closed tracheal suction systems (CTSS) allow suction without having to disconnect the patient. Its main advantages are in reducing the risk of contamination of the environment and health personnel, maintaining positive end-expiratory pressure (PEEP) and lung volume, preventing deoxygenation, and minimizing hemodynamic effects, therefore being very useful in critically ill patients. In addition to these advantages, closed tracheal suction systems can play an important role in preventing VAP [49].

Several studies have been carried out to evaluate the effectiveness of CTSS in preventing VAP [49,50,51]. The most recent meta-analysis, with 16 studies and including almost 2000 patients, showed that the use of closed systems was associated with a reduction in the incidence of VAP (RR 0.69; 95% CI 0.54–0.87; Q = 26.14; I2 = 46.4 %). Despite this reduction in VAP, there was no significant reduction in mortality or time on MV. However, the role of these systems in the prevention of VAP cannot be clearly defined due to the poor quality of the studies included in the analysis [50].

## 10. Discussion

Endotracheal tube colonization by bacteria and biofilm formation, oropharyngeal colonization, and episodes of microaspiration are the main mechanisms for the pathogenesis of VAP. Reducing these mechanisms is important in reducing the incidence of VAP. The development in recent decades of the airway and respiratory devices that interfere with these mechanisms was important in reducing the number of cases of VAP, resulting in improved morbidity in ventilated patients and reducing the impact that this infection has on health costs [4,26].

SDD has been the most studied and apparently most effective system in reducing the incidence of VAP. The results obtained have led to the recommendation of its use, either alone or with other preventive measures for VAP, such as head-of-bed elevation and SDD, to reduce the incidence of VAP. Its use has been shown to be safe and the demonstration of good cost-effectiveness should lead to its more widespread use, especially in patients with a predictably longer mechanical ventilation time.

Coated endotracheal tubes have also been shown to prevent the colonization of ETT and reduce the incidence of VAP, but their main disadvantage is the loss of efficacy over time. Thus, in patients with prolonged mechanical ventilation, in order to maintain the preventive effect on biofilm formation, a programmed replacement of the ETT would be necessary, which could increase the risk of a failed extubation and/or increase the associated costs.

Other systems, such as the CCPC and the CTSS, have been shown to reduce the incidence of VAP, but clinical data are still scarce or of too low a quality to issue definitive recommendations on their use in the prevention of VAP.

On the other hand, practices such as the use of HME and programmed ventilator circuit change have shown no effect in reducing VAP, even increasing the incidence of infection in some cases, which is why they should be avoided in mechanically ventilated patients (Table 1) [8,48].

To implement the use of these airway and respiratory devices in VAP prevention strategies, their effect on reducing the incidence, cost-effectiveness, and value in multimodal prevention strategies must be evaluated (Figure 1). Rarely has the single use of these devices shown a significant effect in reducing VAP, so their combination with less expensive interventions such as semirecumbent positioning or oral hygiene can be a more effective strategy [17].

## 11. Conclusions

In conclusion, SSD is the only system recommended in the guidelines as an additional practice in the prevention of VAP. Other devices such as continuous control of cuff pressure, silver-coated tracheal tubes, low-volume low-pressure tracheal tubes, closed tracheal suctioning system and the Mucus Shaver showed good results; however, the small number of patients or inconsistent methodologies did not allow robust results to be obtained to sufficiently clearly define their real impact on reducing the incidence of VAP. Devices such as the polyurethane-cuffed ETT and tapered-shape cuff, and scheduled ventilator circuit change do not show significant efficacy in preventing VAP, so their use is not recommended.

## Figures and Tables

**Figure 1 medicina-59-00199-f001:**
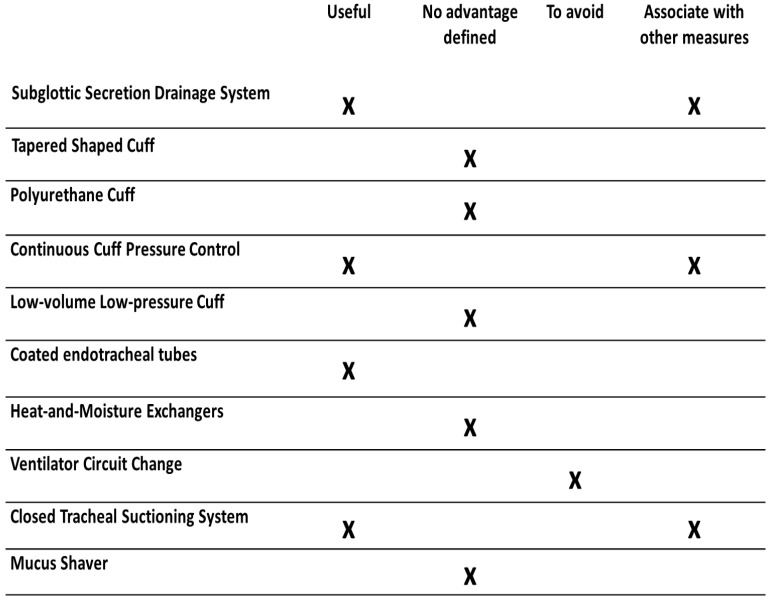
The usefulness of airway and respiratory devices in the prevention of ventilator-associated pneumonia.

**Table 1 medicina-59-00199-t001:** Main characteristics of airway and respiratory devices.

Devices	Advantages	Limitations
Subglottic Secretion Drainage System	Reduced incidence of VAP. Recommended in VAP prevention guidelines.	No reduction in the duration of MV, ICU length of stay, or mortality.
Tapered-Shaped Cuff	No advantages.	No reduction in the incidence of VAP, duration of MV, ICU length of stay, or mortality.
Polyurethane Cuff	Low evidence in the prevention of VAP. Additional data required.	Inaccurate cuff pressure measurement due to cuff condensation.
Continuous Cuff Pressure Control	Reduced risk of VAP, duration of MV, or ICU length of stay.	No effect on mortality. Cost-effectiveness not evaluated. No definite recommendation.
Low-volume Low-pressure Cuff	Reduction of microaspiration. Lower VAP rates in small clinical trials.	Evidence from small clinical trials. No definite recommendation.
Coated endotracheal tubes	Less bacterial colonization. Lower VAP rates.	Effectiveness decreases over time on MV. No definite recommendation.
Heat and Moisture Exchangers	No advantages.	No reduction in the incidence of VAP. Associated with a higher risk of atelectasis and increased dead space. Not recommended.
Ventilator Circuit Change	No advantages.	Scheduled change associated with risk of VAP. Not recommended.
Closed Tracheal Suctioning System	Lower VAP rates.	No reduction in the duration of MV, ICU length of stay, or mortality. No definite recommendation.
Mucus Shaver	Allows the removal of mucus and secretions from inside the ETT.	No impact on ETT colonization. No reduction in the incidence of VAP.

ETT: endotracheal tube, ICU: Intensive Care Unit, MV: mechanical ventilation, VAP: ventilator-associated pneumonia.

## Data Availability

Not applicable.

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
