# Peer review of "Airway and Respiratory Devices in the Prevention of Ventilator-Associated Pneumonia"

_medicina, 2023, doi:10.3390/medicina59020199_

Round 1

Reviewer 1 Report

The authors submitted a review article in which they summarized airway and respiratory devices in the prevention of ventilator-associated pneumonia. The topic is important and the manuscript is well written. However, unfortunately, most of the devices shown negative advantage to patient’s outcomes. The followings are my comment to improve the manuscript.

Major comments

1. The authors focused on airway and respiratory devices to prevent VAP. Recently, multimodal approach has been proven effectiveness to prevent VAP. (PMID: 29023261). I would like to for the authors summarize clinically effective or proven methods which could possibly improve patients’ outcomes. The current article does not seem to provide additional information than guidelines.

2. Introduction

The introduction needs to be reconsidered. These terms, such as VA-LRTI and VAT, do not appear in the latter sentences. This paper focuses on VAP; thus, the introduction also should focus on VAP.

Minor comments

1.

Some of the devices the authors mentioned in the text are not “airway devices”. Thus, “airway and respiratory devices” might be better for the terminology.

2. P2 L62-63

The 2022 61 American practice recommendations of the Society for Healthcare Epidemiology in collaboration with the Infectious Diseases Society of America reclassified SSD from an essential practice to an additional approach to VAP prevention.”

→Citations are needed.

Author Response

The authors submitted a review article in which they summarized airway and respiratory devices in the prevention of ventilator-associated pneumonia. The topic is important and the manuscript is well written. However, unfortunately, most of the devices shown negative advantage to patient’s outcomes. The followings are my comment to improve the manuscript.

Major comments

  1. The authors focused on airway and respiratory devices to prevent VAP. Recently, multimodal approach has been proven effectiveness to prevent VAP. (PMID: 29023261). I would like to for the authors summarize clinically effective or proven methods which could possibly improve patients’ outcomes. The current article does not seem to provide additional information than guidelines.

R: Many thanks for your comments. We fully agree with the reviewer. We discussed the issue in the conclusions according to the suggestions of the reviewer.

  1. Introduction. The introduction needs to be reconsidered. These terms, such as VA-LRTI and VAT, do not appear in the latter sentences. This paper focuses on VAP; thus, the introduction also should focus on VAP.

R: Many thanks for your comments. We fully agree with the reviewer. We rewrote the Introduction according to the suggestion of the reviewer and removed the terms VA-LRTI and VAT.

Minor comments

  1. Some of the devices the authors mentioned in the text are not “airway devices”. Thus, “airway and respiratory devices” might be better for the terminology.

R: Many thanks for your comments. We fully agree with the Reviewer. We add modified the term ‘’airway devices’’ to ‘’airway and respiratory devices’’ in the title, abstract, main text, and table 1.

  1. P2 L62-63 “The 2022 61 American practice recommendations of the Society for Healthcare Epidemiology in collaboration with the Infectious Diseases Society of America reclassified SSD from an essential practice to an additional approach to VAP prevention.” →Citations are needed.

R: Many thanks for your comments. We missed that reference. We added the reference Klompas et al, Infect Control Hosp Epidemiol. 2022;43(6):687-713. doi:10.1017/ice.2022.88 to the text. 

Reviewer 2 Report

This review is comprehensive, so I have minor suggestions.

1.  Please add more table or figure to compare and show these device.

2.  Please check the format according to the author instruction

Author Response

This review is comprehensive, so I have minor suggestions.

  1. Please add more table or figure to compare and show these device.

R: Many thanks for your comments. We created a new figure showing the usefulness of the various systems in the prevention of VAP.

  1. Please check the format according to the author instruction

R: Many thanks for your comments. We add the back matter information missed in the previous version.

Round 2

Reviewer 1 Report

Thank you for revising the manuscript. The authors answered all of my comments successfully, and manuscript has improved. I believe this narrative review would be informative for clinicians.